# Hyperphosphatemia Contributes to Skeletal Muscle Atrophy in Mice

**DOI:** 10.3390/ijms25179308

**Published:** 2024-08-28

**Authors:** Kylie Heitman, Seth Bollenbecker, Jordan Bradley, Brian Czaya, Abul Fajol, Sarah Madison Thomas, Qing Li, Svetlana Komarova, Stefanie Krick, Glenn C. Rowe, Matthew S. Alexander, Christian Faul

**Affiliations:** 1Division of Nephrology and Section of Mineral Metabolism, Department of Medicine, Heersink School of Medicine, The University of Alabama at Birmingham, Birmingham, AL 35294, USA; krh16@uab.edu (K.H.); jb788@evansville.edu (J.B.); bczaya@mednet.ucla.edu (B.C.); afajol@uabmc.edu (A.F.); smthoma2@uab.edu (S.M.T.); qli2@uab.edu (Q.L.); svetlanakomarova@uabmc.edu (S.K.); 2Division of Pulmonary, Allergy and Critical Care Medicine, Department of Medicine, Heersink School of Medicine, The University of Alabama at Birmingham, Birmingham, AL 35294, USA; setheb@uab.edu (S.B.); skrick@uabmc.edu (S.K.); 3Division of Cardiovascular Disease, Department of Medicine, Heersink School of Medicine, The University of Alabama at Birmingham, Birmingham, AL 35294, USA; glennrowe@uabmc.edu; 4Center for Exercise Medicine, The University of Alabama at Birmingham, Birmingham, AL 35294, USA; matthewalexander@uabmc.edu; 5Division of Neurology, Department of Pediatrics, Children’s of Alabama, The University of Alabama at Birmingham, Birmingham, AL 35233, USA; 6Department of Genetics, The University of Alabama at Birmingham, Birmingham, AL 35294, USA; 7Civitan International Research Center, The University of Alabama at Birmingham, Birmingham, AL 35294, USA; 8Center for Neurodegeneration and Experimental Therapeutics, The University of Alabama at Birmingham, Birmingham, AL 35294, USA

**Keywords:** chronic kidney disease, hyperphosphatemia, phosphate, sarcopenia, skeletal muscle atrophy

## Abstract

Chronic kidney disease (CKD) is associated with various pathologic changes, including elevations in serum phosphate levels (hyperphosphatemia), vascular calcification, and skeletal muscle atrophy. Elevated phosphate can damage vascular smooth muscle cells and cause vascular calcification. Here, we determined whether high phosphate can also affect skeletal muscle cells and whether hyperphosphatemia, in the context of CKD or by itself, is associated with skeletal muscle atrophy. As models of hyperphosphatemia with CKD, we studied mice receiving an adenine-rich diet for 14 weeks and mice with deletion of Collagen 4a3 (*Col4a3*^−/−^). As models of hyperphosphatemia without CKD, we analyzed mice receiving a high-phosphate diet for three and six months as well as a genetic model for klotho deficiency (*kl*/*kl*). We found that adenine, *Col4a3*^−/−^, and *kl*/*kl* mice have reduced skeletal muscle mass and function and develop atrophy. Mice on a high-phosphate diet for six months also had lower skeletal muscle mass and function but no significant signs of atrophy, indicating less severe damage compared with the other three models. To determine the potential direct actions of phosphate on skeletal muscle, we cultured primary mouse myotubes in high phosphate concentrations, and we detected the induction of atrophy. We conclude that in experimental mouse models, hyperphosphatemia is sufficient to induce skeletal muscle atrophy and that, among various other factors, elevated phosphate levels might contribute to skeletal muscle injury in CKD.

## 1. Introduction

Chronic kidney disease (CKD) is a public health epidemic that affects an estimated 26 million Americans and more than 800 million individuals worldwide [1,2]. CKD is associated with various pathologies, including vascular calcification, pathologic cardiac remodeling, systemic inflammation, and anemia, which together contribute to the high mortality rates [3,4,5,6]. Furthermore, about 2/3 of patients with CKD progressively lose skeletal muscle mass and strength, a condition called sarcopenia, which results in frailty and a decline in physical performance [7]. Sarcopenia is associated with CKD severity and increased mortality in non-dialysis and dialysis CKD patients, and markers of muscle mass and strength are important predictors of poor outcomes in all stages of CKD [7].

In sarcopenia, skeletal muscle tissue loses its protein stores, also referred to as muscle protein catabolism or wasting, which is attributed to a disruption in overall protein balance caused by the suppression of protein synthesis, the stimulation of protein degradation, and the impaired growth of new muscle fibers. The persistence of muscle wasting results in a thinning of myofibers, also called atrophy, and a loss of muscle mass, leading to reduced muscle function. Although CKD patients lose overall body weight, it is the reduction in skeletal muscle mass that is associated with increased mortality [8]. Besides the atrophy of myofibers, CKD-associated sarcopenia also includes changes in other cell types and cellular processes in skeletal muscle tissue, such as interstitial fibrosis and the infiltration of inflammatory cells [9,10,11,12]. Studies in CKD patients have shown that not only a decrease in muscle mass but also in muscle quality is associated with low muscle strength, reduced physical performance, and increased mortality [13,14,15].

Although clinical studies provide strong evidence that sarcopenia is a major pathology that is associated with CKD, predicts poor outcomes, and contributes to high mortality, our understanding of the underlying causes and mechanisms is still in its infancy [7]. While insufficient food intake due to anorexia and dietary restrictions contribute to muscle loss, several features of CKD-associated sarcopenia cannot be explained by an inadequate diet alone. Similarly, sarcopenia is not simply a consequence of the low physical activity of CKD patients. Instead, sarcopenia is part of a disease process that is associated with a catabolic state, oxidative stress, systemic inflammation, and impaired insulin signaling [16,17,18,19,20,21,22]. While CKD and sarcopenia share common risk factors, such as diabetes, obesity, and aging, CKD is also accompanied by alterations in specific factors that can directly or indirectly affect skeletal muscle tissue and contribute to sarcopenia, including metabolic acidosis and the accumulation of uremic toxins [12,23,24,25,26,27]. Current therapies include exercise and nutritional management, but specific pharmacological treatments for preventing sarcopenia in CKD are not available.

Surprisingly, compared with other pathologies, such as cardiovascular diseases, sarcopenia has not been studied in great detail in animal models of CKD [7]. Here, we provide a comprehensive analysis of pathologic changes in skeletal muscle on a histological, cellular, and molecular level in two mouse models of CKD, which should help to uncover causative pathomechanisms and thereby identify novel therapeutic targets. Alterations in mineral metabolism are a hallmark of CKD, which includes increases in systemic phosphate levels, also called hyperphosphatemia, as well as elevations in serum levels of fibroblast growth factor 23 (FGF23), which is a bone-derived phosphaturic hormone [28]. It has been shown that hyperphosphatemia can contribute to the progression of kidney injury [29] and to the damage of other tissues, such as vascular calcification and cardiac hypertrophy, as well as to premature death [30]. Therefore, lowering systemic elevations of phosphate has therapeutic potential in CKD [31]. To determine whether high phosphate by itself can cause sarcopenia, we also analyzed skeletal muscle tissue in two mouse models with hyperphosphatemia in the absence of CKD, and we studied the direct effects of elevated phosphate concentrations on skeletal muscle cells in culture.

## 2. Results

### 2.1. Mice Fed an Adenine-Rich Diet Develop Skeletal Muscle Atrophy

Mice receiving an adenine-enriched diet for several weeks develop rapid and progressive CKD [32]. The accumulation of adenine-containing crystals in the kidney causes tubular atrophy and tubulointerstitial fibrosis, and animals develop cardiovascular pathologies, including vascular calcification and cardiac hypertrophy [33,34]. To determine if this CKD model also develops pathologic changes in skeletal muscle, we placed eight-week-old C57BL/6J mice on an adenine-rich diet for 14 weeks. Levels of blood urea nitrogen (BUN) (Figure 1A) as well as serum levels of phosphate (Figure 1B) and FGF23 (Figure 1C) were significantly elevated compared with mice receiving a normal diet. Magnetic resonance imaging (MRI) of the hindlimb showed that mice on an adenine-rich diet had a decreased cross-sectional muscle area (Figure 1D,E). These changes were accompanied by significant reductions in grip strength (Figure 1F) and in muscle mass (Figure 1G). Furthermore, histological analysis of the gastrocnemius muscle that was immunostained with anti-laminin revealed a significant decrease in the cross-sectional area of individual myofibers (Figure 1H,I). We also analyzed skeletal muscle tissue on a molecular level. Atrophy is driven by increased proteolysis, which is controlled by the ubiquitin-proteasome system [35]. Atrogenes, such as atrophy-related muscle-specific E3 ligases, including muscle-specific RING finger protein 1 (MuRF1, also called tripartite motif containing 63 or TRIM63) and Atrogen1 (also called muscle-atrophy F-box protein or FBXO32), augment protein degradation, and increases in their expression levels serve as markers of atrophy [36]. Myostatin, also known as growth differentiation factor 8 (GDF-8), is a myokine that belongs to the transforming growth factor (TGF) β subfamily [37]. Myostatin acts in a paracrine fashion on myofibers and suppresses the growth of skeletal muscle by inducing protein degradation and thereby atrophy [38]. Metallothioneins (MT) belong to a family of cysteine-rich, metal-binding proteins that play a role in cellular zinc homeostasis, mitochondrial function, and defense against oxidative stress [39]. MT-1 expression is elevated in skeletal muscle atrophy [40]. Our quantitative real-time polymerase chain reaction (qRT-PCR) analysis showed that the expression of *Trim63*, *Fbxo32*, *Myostatin*, and *MT1* was significantly elevated in the gastrocnemius muscle of mice receiving an adenine-rich diet when compared with mice on a normal diet (Figure 1J–M). Interestingly, we did not detect significant elevations in other types of skeletal muscle, including the quadriceps, soleus, and tibialis anterior.

In sarcopenia, muscle atrophy is often accompanied by inflammation and interstitial fibrosis [41]. However, when we determined expression levels of the pro-inflammatory cytokines *Interleukin* (*Il*) *6* and *Tumor necrosis factor* (*Tnf*) *s* by qRT-PCR analysis in different types of skeletal muscle, we could not detect changes between mice on adenine-rich and normal diets (Appendix A). Similarly, histological analysis of collagen content by Picrosirius red staining (Appendix A) as well as qRT-PCR expression analysis of the pro-fibrotic markers *Tgfβ*, *Collagen type III alpha 1 chain* (*Col3a1*), and *Fibronectin* showed no differences between both groups (Appendix A). Combined, our findings indicate that mice on an adenine-rich diet develop kidney damage and hyperphosphatemia, which is accompanied by skeletal muscle atrophy and reductions in skeletal muscle mass and function. Myofiber atrophy seems to occur in a muscle-type-dependent manner and is not accompanied by interstitial fibrosis or inflammation.

### 2.2. Col4a3^−/−^ Mice Develop Skeletal Muscle Atrophy

Mice with global deletion of the *collagen type IV alpha 3 chain* (*Col4a3*^−/−^) develop glomerular injury that progresses to CKD and are a model of Alport syndrome [42]. We evaluated the skeletal muscle pathologies in *Col4a3*^−/−^ mice at ten weeks of age, when kidney injury is severe [43]. As expected, levels of BUN (Figure 2A) as well as serum phosphate (Figure 2B) and FGF23 (Figure 2C) were significantly elevated when compared with wildtype littermates. Hindlimb area (Figure 2D,E), grip strength (Figure 2F), and muscle mass (Figure 2G) were significantly reduced in *Col4a3*^−/−^ mice. The cross-sectional area of individual myofibers was significantly decreased (Figure 2H,I), which was accompanied by significant elevations in the expression levels of *Trim63*, *Fbxo32*, *Myostatin*, and *MT1* (Figure 2J–M). Immunoblot analysis of protein extracts from the gastrocnemius muscle confirmed increases in TRIM63 and FBXO32 expression on protein level in *Col4a3*^−/−^ mice (Figure 2N–P). The analysis of expression levels of *Il6* and *Tnfα* by qRT-PCR (Appendix A) and of collagen content by Picrosirius red staining (Appendix A) as well as qRT-PCR expression analysis of *Tgfβ* (Appendix A) showed no significant differences between *Col4a3*^−/−^ mice and wildtype controls. Expression levels of *Col3a1* were significantly reduced in *Col4a3*^−/−^ mice (Appendix A).

It is known that at five weeks of age, *Col4a3*^−/−^ mice do not show significant kidney damage or elevations in serum levels of phosphate and FGF23 [43]. Here, we could also not detect significant differences in grip strength (Figure 2Q) or skeletal muscle mass (Figure 2R) between five-week-old *Col4a3*^−/−^ mice and wildtype littermates. Additionally, expression levels of *Trim63*, *Fbxo32*, *Myostatin*, and *MT1* were similar in both groups (Figure 2S–V). Combined, our findings indicate that *Col4a3*^−/−^ mice develop skeletal muscle atrophy that is not accompanied by interstitial fibrosis or inflammation. In this model, pathologic changes in skeletal muscle seem to only occur when the kidney is damaged and when serum phosphate levels are elevated.

### 2.3. Klotho-Deficient Mice Develop Skeletal Muscle Atrophy

*Klotho*-deficient mice are a genetic model of hyperphosphatemia without severe kidney injury [44]. Homozygous mice (*kl*/*kl*) develop a phenotype of accelerated aging and die by ten weeks of age. When we analyzed the skeletal muscle phenotype in eight-week-old *kl*/*kl* mice, we found that the hindlimb area (Figure 3A,B), grip strength (Figure 3C), and muscle mass (Figure 3D) were significantly reduced. Furthermore, the cross-sectional area of individual myofibers was significantly decreased (Figure 3E,F), which was accompanied by significant elevations in the expression levels of *Trim63*, *Fbxo32*, *Myostatin*, and *MT1* (Figure 3G–J). Immunoblot analysis of the gastrocnemius muscle confirmed increases in TRIM63 and FBXO32 expression on protein level in *kl*/*kl* mice (Figure 3K–M).

The analysis of mRNA levels of *Il6* by qRT-PCR (Appendix A) and of collagen content by Picrosirius red staining (Appendix A), as well as qRT-PCR expression analysis of *Tgfβ*, *Col3a1*, and *Fibronectin* (Appendix A), showed no significant differences between *kl*/*kl* mice and wildtype controls. Our study in *kl*/*kl* mice suggests that skeletal muscle atrophy and reductions in skeletal muscle mass and function can occur in the context of hyperphosphatemia in the absence of CKD. In this scenario, myofiber atrophy does not seem to be accompanied by interstitial fibrosis or inflammation.

### 2.4. Mice Fed a High-Phosphate Diet Have Reduced Skeletal Muscle Mass and Strength

To induce hyperphosphatemia in the absence of severe kidney disease, we administered a 3% high-phosphate diet to C57BL/6J mice for three and six months [5,45]. BUN levels were not elevated (Figure 4A), and increases in serum levels of phosphate (Figure 4B) and FGF23 (Figure 4C) were significant after six months of a high-phosphate diet when compared with C57BL/6J mice receiving a normal diet for six months. When we analyzed hindlimb area, grip strength, and muscle mass after three months of a high-phosphate diet, we could not detect significant changes between mice on both diets (Figure 4D–G). However, after six months. grip strength and muscle mass were reduced (Figure 4F,G), as previously reported [5]. Interestingly, these changes were not accompanied by decreases in the cross-sectional area of individual myofibers (Figure 4D,I) or by elevations in the expression levels of *Trim63* or *Fbxo32* (Figure 4J,K). However, the expression of *Myostatin* showed a trend to increase (Figure 4L), and *MT1* expression was significantly elevated (Figure 4M).

The analysis of expression levels of *Il6* by qRT-PCR (Appendix A) and of collagen content by Picrosirius red staining (Appendix A), as well as the qRT-PCR expression analysis of *Tgfβ*, *Col3a1*, and *Fibronectin* (Appendix A), showed no significant differences in mice receiving a high-phosphate diet for three or six months when compared with mice on a normal diet. Our study suggests that dietary phosphate overload in healthy mice reduces skeletal muscle mass and strength over time. However, this skeletal muscle pathology does not seem to be driven by myofiber atrophy, interstitial fibrosis, or inflammation.

### 2.5. Phosphate Elevations Induce Atrophy in Cultured Myotubes

Since our in vivo studies indicate that hyperphosphatemia itself might cause skeletal muscle damage, we analyzed the effects of phosphate elevations on cultured muscle cells. We isolated myotubes from mice, treated them with increasing concentrations of phosphate for 24 h, and analyzed the expression levels of atrogenes. Dexamethasone was used as a positive control for the induction of atrophy [46,47], and Na_2_SO_4_ was used as an anion-negative control [5]. We detected increases in the expression levels of *Trim63*, *Fbxo32*, *Myostatin*, and *MT1* in the presence of 3 mM phosphate by qRT-PCR analysis (Figure 5A–D). However, when myotubes were co-treated with phosphonoformic acid (PFA), which is an inhibitor of phosphate transporters [48], phosphate-induced changes were not observed (Figure 5E–H). Similarly, co-treatment with an inhibitor of FGFRs, which can act as phosphate sensors on the cell surface [49], reduced the effects of 4 mM phosphate (Figure 5E–H). We also studied the atrophy of primary mouse myotubes via microscopic analysis of cell dimensions. We found that treatment with 4 mM phosphate for 24 h significantly reduced the length and area of individual myotubes, which did not occur when cells were co-treated with PFA (Figure 5I–K). Our in vitro studies suggest that elevated phosphate can directly induce myotube atrophy, which might involve the sensing and uptake of phosphate.

### 2.6. FGF23 Does Not Directly Affect Skeletal Muscle

We have previously shown that FGF23 can directly target cardiac myocytes and that elevated serum levels of FGF23 can induce cardiac hypertrophy in animal models of CKD [50]. Since all mouse models studied here also develop increased serum FGF23 concentrations, we wanted to determine whether FGF23 can directly affect skeletal muscle. C2C12 myoblasts and myotubes, as well as mouse skeletal muscle tissue, express FGFRs and klotho (Appendix A), as also shown by others [51], and therefore they have the molecular make-up to respond to FGF23. However, treatment with FGF23 did not increase proliferation of C2C12 myoblasts (Appendix A) or induce the expression of atrogenes in C2C12 myotubes (Appendix A), while FGF2 [52] and dexamethasone, used as respective positive controls, had such effects. FGF23 induces Ras/mitogen-activated protein kinase (MAPK) signaling in target tissues that express FGFRs and klotho, such as the kidney [53]. When we injected recombinant FGF23 protein into wildtype mice and isolated tissues 20 min later, we could detect increased phosphorylation of ERK1/2 in the kidney by immunoblot analysis, indicating Ras/MAPK activation (Appendix A). However, phospho-ERK1/2 levels were not elevated in skeletal muscle tissue when compared with mice receiving solvent instead of FGF23. This finding is in agreement with a previous FGF23 injection study in rats showing that FGF23 does not increase the expression of *Early growth response protein 1* (*EGR1*), another component of Ras/MAPK signaling, in skeletal muscle tissue [54]. Combined, these findings indicate that skeletal muscle cells do not respond to elevations in FGF23 and that high FGF23 might not contribute to skeletal muscle atrophy.

## 3. Discussion

In this study, we found that four mouse models of hyperphosphatemia, two with CKD and two without CKD, develop skeletal muscle injury. Hyperphosphatemia induced by the administration of a high-phosphate diet caused the fewest alterations, suggesting that the degree of phosphate elevation might determine the severity of skeletal muscle injury. Furthermore, we detected skeletal muscle injury in mice after six but not three months of a high-phosphate diet, suggesting that the duration of hyperphosphatemia also affects the degree of skeletal muscle damage. Similarly, *Col4a3*^−/−^ mice only developed skeletal muscle atrophy at ten weeks of age, when kidney function is reduced and phosphate concentrations are elevated, but not at five weeks, when kidneys and phosphate levels are normal. Combined with our previous study showing that ten-week-old *Col4a3*^−/−^ mice receiving a low-phosphate diet exhibit reduced serum phosphate levels and show no signs of skeletal muscle atrophy [5], our current study suggests that hyperphosphatemia is a pathologic factor that, on its own as well as in the context of CKD, might contribute to skeletal muscle atrophy. The direct pathologic effects of phosphate on skeletal muscle are also indicated by our in vitro studies showing that phosphate elevations in cell culture medium induce atrophy in primary mouse myotubes.

Skeletal muscle atrophy is an important aspect of sarcopenia that has been detected in CKD patients with an increasing severity of kidney injury [7]. To study potential causes of CKD-associated atrophy and identify novel drug targets, it is important to determine if animal models of CKD develop skeletal muscle atrophy. To date, the characterization of skeletal muscle injury has only been conducted in a few CKD models and not in much detail. The partial removal of kidney mass by surgery, also called subtotal nephrectomy, in mice and rats is a well-established animal model of CKD, and to date it is the most studied in regards to CKD-associated changes in skeletal muscle tissue [7]. For the first time, our previous [5] and current studies suggest that *Col4a3*^−/−^ mice serve as a genetic CKD model with skeletal muscle atrophy. Furthermore, adenine mice serve as a diet-induced model of CKD-associated atrophy, as suggested by our study and by others [55,56,57,58]. However, in this model, reports of the cross-sectional area of myofibers as a key readout for atrophy have been inconsistent [7]. It is important to note that mammalian skeletal muscle is composed of different myofiber types based on the expression of distinct myosin heavy chains, and myofiber types differ in their contractile and metabolic characteristics, as well as in their response to the same stimulus and their susceptibility to undergo atrophy. We found in the adenine model that skeletal muscle types have significant differences in their atrophy response, with the gastrocnemius muscle showing the most consistent elevations in atrogenes. The difference in the effects of CKD on different muscle tissues might be based on heterogeneous fiber type composition. Overall, our current animal studies suggest that it is important to study atrophy in a fiber-type-specific context, as otherwise changes might be missed and conclusions might be misleading. Furthermore, while atrophy is consistently detected in the skeletal muscle tissue of CKD patients and rodent models of CKD, only a few human and animal studies have reported skeletal muscle fibrosis and inflammation, and findings have been inconsistent [7]. Our study in adenine and *Col4a3*^−/−^ mice did not detect fibrosis or inflammation in skeletal muscle tissue, even when analyzing different types of skeletal muscle, suggesting that at least in these two CKD models, atrophy and lower muscle mass, rather than reduced muscle quality, are the major drivers of impaired muscle function.

Of note, not all in vivo studies found that animal models of CKD develop skeletal muscle damage [7]. A recent study reported that mice receiving an adenine-rich diet develop reduced skeletal muscle mass only early on, which reverses over time when a normal diet is introduced [59]. Furthermore, this study found that following subtotal nephrectomy, mice show reductions in muscle mass early on due to reduced food intake and a drop in overall body mass; however, later on, mice recover from their pathologic changes in skeletal muscle tissue. Both models suggest that kidney injury per se, which in these mice does not improve over time, might not be a driver of skeletal muscle damage in CKD. While the reasons for discrepancies between this study, the findings of others, and our current study are not clear, they might be based on differences in the precise design of the model and study. Nevertheless, it is important to report all of these findings to demonstrate that animal studies need to be taken with caution when interpreting pathologies and that nuances in the design of the model and study might affect outcomes. The key question that remains is which models reflect the human pathology the best, which will be challenging to answer as the characterization of CKD-associated skeletal muscle injury in patients is even less detailed and not very advanced.

Overall, our in vivo and in vitro studies indicate that elevated phosphate levels can directly, in the presence as well as in the absence of CKD, induce skeletal muscle atrophy. Besides our study, there is growing experimental evidence suggesting that abnormal elevations in extracellular phosphate might be harmful for skeletal muscle. For example, it has been shown that the administration of a high-phosphate diet worsens skeletal muscle atrophy in nephrectomized rats [60,61]. Additionally, dietary phosphate load in wildtype mice does not only induce atrophy, as we have shown here and before [5], but also alters gene expression in skeletal muscle tissue, with elevations in genes regulating glucose metabolism and decreases in genes involved in fatty acid metabolism, which are accompanied by reductions in spontaneous and exercise activities [62]. The findings by us and others that *kl*/*kl* mice develop skeletal muscle atrophy and wasting [63,64,65,66,67,68] and that deletion of the renal phosphate transporter NaPi-2a in *kl*/*kl* mice reduces serum phosphate levels and protects from skeletal muscle atrophy [63] suggest that pathologic changes are not directly caused by the absence of klotho but indirectly by other pathologic changes that occur in the absence of klotho, such as hyperphosphatemia [63]. Furthermore, not only mice with genetic deletion of klotho but also of FGF23 with normal klotho levels develop hyperphosphatemia in the absence of severe kidney damage that is accompanied by skeletal muscle wasting and atrophy [69,70]. While in vitro studies suggest direct pathologic actions of elevated phosphate on skeletal muscle cells [7], the underlying mechanisms remain unclear. Our in vitro experiments suggest that phosphate-induced atrophy might involve phosphate sensing by FGFRs and phosphate uptake by transporters.

Currently, it cannot be excluded that other factors that are involved in the regulation of phosphate metabolism and that are affected by systemic phosphate elevations contribute to the observed atrophy in mice. Such changes include systemic elevations of FGF23 and PTH and reductions in active vitamin D and soluble klotho, which all could have a pathologic impact on skeletal muscle tissue, as indicated by animal studies [7]. However, at least for FGF23, in vitro experiments reported here as well as in a previous study by others [51] suggests that even at high concentrations, FGF23 does not affect myotubes [7]. Nevertheless, it will be challenging to experimentally determine the direct pathologic effects of extracellular phosphate on skeletal muscle, which would require the block of phosphate uptake specifically into myofibers by deleting phosphate transporters. However, since constant phosphate uptake is essential for normal cell function and tissue health, such genetic deletions should have delirious effects. In fact, it has been shown that combined deletion of PiT-1 and PiT-2, two major transporters that mediate phosphate uptake for various housekeeping functions, in skeletal muscle tissue reduces muscle function and impairs survival in mice [71].

To date, only a few human studies have analyzed potential associations between phosphate levels and skeletal muscle injury in CKD or in general [7]. Future clinical studies should determine potential associations between phosphate and sarcopenia in CKD, especially in dialysis patients. Many of these patients receive phosphate-lowering therapies, such as a phosphate-restricted diet or phosphate binders, which could have protective effects not only on the cardiovascular system but also on skeletal muscle and thereby prolong survival. Furthermore, phosphate levels rise during aging, and it has been shown that in aged mice, elevated serum phosphate concentrations are associated with reduced muscle strength [72,73]. Furthermore, aged mice receiving a low-phosphate diet show improved muscle function, with a larger myofiber area and increased muscle strength and physical performance [73,74]. It would be interesting to determine whether hyperphosphatemia is associated with sarcopenia in the elderly. Finally, our high-phosphate feeding study in healthy mice suggests that the frequent intake of processed food that is high in phosphate-containing salt content might cause skeletal muscle damage. Overall, our experimental findings indicate that hyperphosphatemia on its own and in the context of CKD might contribute to sarcopenia. Future studies need to determine the effects of phosphate in comparison to the many other pathologic changes that occur in the context of hyperphosphatemia, as well as the clinical relevance of our experimental findings.

## 4. Materials and Methods

### 4.1. Mice

All animal studies were conducted according to applicable laws and guidelines and were approved by the Institutional Animal Care and Use Committees (IACUC) at the University of Alabama Birmingham (UAB) School of Medicine. Studies were performed using female and male mice, and mice were maintained on a NIH31 rodent diet (Harlan Teklad) and fed ad libitum, unless otherwise indicated. Constitutive *Col4a3*^−/−^ (Alport) mice [42] and *kl*/*kl* mice [44] were maintained on a mixed Sv129/C57BL/6 background in a heterozygous breeding state. Both mouse models were housed in our UAB rodent facility with 12 h light/dark cycle. Mice had free access to regular water and chow. *Col4a3*^−/−^, *kl*/*kl*, and adenine mice were also provided with HydroGel water packs (Clear H_2_O; 70075022). *Col4a3*^−/−^ mice were euthanized at the age of five or ten weeks, *kl*/*kl* mice at eight weeks. Respective wildtype littermates served as controls.

For the adenine model, male C57BL/6J mice were purchased from Jackson Laboratory, and after a 1-week acclimation period, mice were placed on a 14-week adenine diet: 6 weeks of a 0.2% adenine diet (TD.140290), followed by 2 weeks of a 0.15% adenine diet (TD.170304), and an additional 6 weeks of 0.2% adenine diet. Male C57BL/6J mice receiving a control adenine diet (TD.170303) for 14 weeks served as controls. For high-phosphate feeding studies, male and female C57BL/6J mice were purchased from Jackson Laboratory and placed on a 3% high-phosphate diet (TD.180286) or a control diet (TD.180287). All mice were euthanized with 2.5% isoflurane.

### 4.2. Recombinant Proteins, Antibodies, and Small Molecules

Recombinant proteins from R&D Systems were mouse FGF1 (4686FA025/CF), mouse FGF2 (3139FB025CF), and mouse FGF23 (2629FG025/CF). Primary antibodies used were against α-actinin (AA7732, Sigma-Aldrich, St. Louis, MO, USA); laminin (L9393, Sigma-Aldrich); MAFbx (166806, Santa Cruz Biotechnology, Dallas, TX, USA); MuRF1 (398608, Santa Cruz Biotechnology); total ERK (4695S, Cell Signaling, Danvers, MA, USA); phosphorylated ERK (9101S, Cell Signaling); and GAPDH (CB1001, Millipore, Burlington, MA, USA). Secondary antibodies were horseradish peroxidase (HRP)-conjugated anti-mouse (W4021, Promega, Madison, WI, USA) and anti-rabbit (W4011, Promega) for Western blotting; and Alexa488-conjugated Alpaca Anti-Rabbit (611545215, Jackson Immuno Research Laboratories, West Grove, PA, USA) and Alexa488-conjugated Donkey Anti-Mouse (715545150, Jackson Immuno Research Laboratories) for immunocytochemistry and immunohistochemistry. We used PD1703744 (Sigma-Aldrich) as a pan-FGFR inhibitor [75] and phosphonoformic acid (PFA; P6801, Sigma-Aldrich) as a PiT-1/2 inhibitor [48].

### 4.3. Magnetic Resonance Imaging (MRI)

MRI images were taken by the Small Animal Imaging Facility at UAB. Mice were imaged with a 9.4 Tesla Bruker Biospec scanner (Bruker Biospin, Billerica, MA, USA) in prone position on an MRI-compatible bed system with circulating heated water. Isoflurane gas was used for anesthesia, and respiration was monitored with an MRI-compatible physiological monitoring system (SA Instruments Inc., Stonybrook, NY, USA). A 72 mm diameter birdcage column coil was used for signal excitation and a 24 mm diameter surface coil for receptor (Doty Scientific Inc., Columbia, SC, USA). A 2D T2-weighted fast spin echo sequence was used for imaging of the mouse left leg calf area in the axial, sagittal, and coronal orientations with and without fat suppression. The following parameters were used: TR/TE = 200/25 ms, ETL = 4, 2 averages, 27 contiguous slices (axial) and 23 slices (coronal) with 1 mm thickness, FOX = 30 × 30 mm, and matrix = 300 × 300 for 100 μM in-plane resolution.

### 4.4. Grip Strength

Before euthanasia, grip strength was measured using a Chatillon DFE series digital force gauge (E-DFE-200, Chatillon, France), provided by the Behavioral Assessment Core at UAB. Mice were placed on a metal grid and pulled backwards by their tails until grip could not be maintained. Each mouse was given 10 trials, excluding the highest and lowest values, and data were reported in Newtons of force (N) as averages normalized to tibia length in mm as measured with a digital caliper (CDS6″CT, Mitutoyo, Kawasaki, Japan).

### 4.5. Tissue Collection

Four muscle groups were isolated from the left leg of mice, including the gastrocnemius, soleus, tibialis anterior, and quadriceps muscles, and snap frozen in liquid nitrogen. The same muscles were isolated from the right leg and placed in Tissue-Tek Cryomold (4557, Sakura, Osaka, Japan) with O.C.T. (4583, Sakura), frozen in 2-methyl-butane, and cooled in liquid nitrogen before being stored at −80 °C.

### 4.6. Serum Chemistry

Blood collection was taken through cardiac puncture, dispensed into serum collection tubes (15.16.74, Sarstedt, Nümbrecht, Germany), and centrifuged at 7000× *g* at room temperature for 10 min. Serum was collected and stored at −80 °C. Serum phosphate and BUN biochemistries were analyzed by IDEXX Bioanalytic Laboratories (Columbia, MO, USA), and serum FGF23 was quantified using an ELISA detecting total mouse FGF23 (60-6300, Quidel, San Diego, CA, USA).

### 4.7. Histology

For immunofluorescence microscopy, cryo-embedded muscle tissues were sectioned using a Leica CM 1950 at 7 μM at −20 °C onto a glass slide (12-550100, Fisher Scientific, Waltham, MA, USA). BSA buffer was used to block tissue sections for 30 min, followed by 1 h incubation with 1:1000 anti-laminin antibody. After, the primary antibody was removed, and sections were washed 3× for 5 min with BSA blocking buffer, followed by the addition of 1:500 Alexa 488 Alpaca anti-rabbit secondary antibody for 1 h, followed by 3× 5-min washes with BSA blocking buffer. Prolong Diamond Antifade Mountant with 4′,6-diamidino-2-phenylindole (DAPI; P36962, Invitrogen, Waltham, MA, USA) was added to tissues before being covered with a glass cover slide. Sections were set to dry in the dark at room temperature overnight before being stored at −80 °C. For Picrosirius red staining, the gastrocnemius muscle was processed and stained at the UAB Comparative Pathology Lab. Immunofluorescence and Picrosirius red sections were imaged using a Leica DMi8 microscope.

### 4.8. RNA Isolation from Tissue and Quantitative Real-Time PCR

Total RNA from frozen muscle tissue or harvested cells was isolated using the RNeasy Plus Universal Mini Kit (73404, Invitrogen). RNA was reverse transcribed into cDNA using iScript supermix (1708840, BioRad, Hercules, CA, USA), and qRT-PCR was performed in duplicates using SYBR Green supermix (172-5272, BioRad) up to 40 cycles (95 °C, 30 s; 98 °C, 15 s; 60 °C, 30 s; 65 °C, 5 s) on a CFX96 Touch Real Time Detection Instrument (1855196, BioRad). Primer sequences are presented in Appendix A. Samples were run in duplicates, and relative gene expression was normalized to GAPDH using 2^−∆∆Ct^ or absolute gene expression was normalized to GAPDH using 2^40−(∆CT)^.

### 4.9. Protein Isolation and Western Blotting

30 mg of the mouse gastrocnemius muscle was homogenized in 5× RIPA lysis buffer (50 uM Tris-HCl, pH 7.5, 200 nM NaCl, 1% Triton X-100, 0.25% deoxycholic acid, 1 mM EDTA, 1 mM ethylene glycol-bis(β-aminoethyl ether)-N,N,N′,N′-tetraacetic acid tetrasodium (EGTA), and a protease and phosphatase inhibitor (11836153001, Roche (Basel, Switzerland) and P5726, P0044, Sigma-Aldrich). Muscle homogenates were incubated on ice for 1 h, followed by a 45 min spin-down at 13,000 g at 4 °C. An amout of 200 μL of protein lysate was transferred into a new Eppendorf tube, following an additional 200 μL of RIPA and an additional 45 min spin-down at 13,000 g at 4 °C. Laemmli buffer (1610747, BioRad) and beta-mercaptoethanol (1610710, BioRad) were added, and samples were vortexed and boiled for 5 min at 95 °C and stored at −80 °C. For protein isolation from cultured cells, 250 μL of RIPA lysis buffer was added to each well of a 6-well plate, and cells were homogenized with a cell lifter (3008, Costar, Washington, DC, USA). Lysates were transferred to an Eppendorf tube, placed on ice for 30 min, and centrifuged at 13,000× *g* at 4 °C for 30 min. Supernatants were transferred into a new Eppendorf tube, Laemmli buffer and beta-mercaptoethanol were added, and samples were vortexed, centrifuged, and boiled for 5 min at 95 °C and stored at −80 °C. An amount of 20 μL of total protein was loaded onto a 12% SDS polyacrylamide gel and separated by SDS-PAGE in 1× Tris/Glycine/SDS buffer (1610732, BioRad) at 0.2 Amp/gel. Proteins were transferred onto PVDF membranes (IPVH00010, Merck Millipore, Burlington, MA, USA) by semi-dry blotting using 1× Tris/Glycine Buffer (1610734, BioRad) and 20% methanol at 23 V for 1 h. Membranes were blocked with 5% non-fat dry milk with 0.1% Tween-20 in 1× Tris buffered saline (TBST) for 1 h. Primary antibodies were added at 1:10,000 for GAPDH and 1:1000 for all other primary antibodies and incubated overnight on a shaker at 4 °C. Membranes were rinsed 3× for 10 min with TBST, followed by the addition of secondary anti-mouse or anti-rabbit antibodies at 1:10,000 in 5% non-fat milk for 1 h at room temperature, followed by 3× 10 min washes in TBST. Membranes were activated using chemiluminescence detection solution (RPN2106, GE Healthcare, Hertfordshire, UK) for 2 min and imaged on an SRX-101A X-ray film developer. Some membranes were activated using Clarity Western ECL Substrate (1705060, BioRad) and imaged using a BioRad ChemiDoc Imaging System (Software Version: 2.1.0.32 BETA CL19032).

### 4.10. Primary Cell Culture

Freshy obtained 2–5 mg of hindlimb skeletal muscle tissue from 3–8-week-old C57BL/6J mice was minced, followed by a 1-h digestion using 2.4 units/mL of neutral protease (Dispase; LS02140, Worthington Biochemical Corp, Washington, DC, USA) and 10 mg/mL of collagenase D (11088882201, Roche Applied Science, Basel, Switzerland) in a shaker at 37 °C in a 50 mL conical tube. After 1 h, 10 mL of growth media (Promocell human skeletal muscle growth media; C-23060 + 1× Glutamax; 35050; Invitrogen) including 1× antibiotic-antimycotic (15240, Invitrogen) and 20% fetal bovine serum (26140079, Gibco) was filtered through a 70 μM filter (352350, Falcon), and centrifuged for 10 min at 1600 rpm at 9 °C. After removal of the supernatant, the cell pellet was resuspended in 3 mL of red blood cell lysis buffer (1589904, Qiagen, Germantown, MD, USA) for 5 min, before the addition of 22 mL of growth media to stop the lysis reaction. The solution was run through a 40 mL filter (431750, Corning, Corning, NY, USA) and centrifuged for 10 min at 1600 rpm at 9 °C. The cell pellet was resuspended in growth media and plated in a sterile 10 cm dish (353003, Falcon) for 1 h before transferring the cell media to a 10 cm dish that was pre-coated with 0.1% gelatin (07903, Stemcell Technologies, Vancouver, BC, Canada). The cells were split and pre-plated 5–15 more times, until reaching a pure culture of myoblasts. Cells were split and plated into 6-well plates (FB01297, Fisherbrand, Waltham, MA, USA), or cells were seeded onto chamber slides (354104, Falcon, Warren, NJ, USA) coated with 0.1% gelatin. Once cells reached 80% confluency, media was switched to differentiation media (Promocell human skeletal muscle differentiation media, C-23061 + 1× Glutamax; 35050; Invitrogen), including 1× antibiotic antimycotic (15240062, Gibco) and 2% horse serum (26050-088, Gibco, Waltham, MA, USA) to initiate myoblast-to-myotube differentiation.

For phosphate treatment studies, sodium phosphate dibasic anhydrous (Na_2_HPO_4_) (BP332-500, Fisher Scientific, Waltham, MA, USA) and sodium phosphate monobasic anhydrous (NaH_2_PO_4_) (BP329-1, Fisher Scientific) were used to prepare a 0.5M stock sodium phosphate buffer solution containing 250 mM Na_2_HPO_4_ and 250 mM NaH_2_PO_4_ at an adjusted pH of 7.4. Cells were treated with 1 or 3 mM of sodium phosphate or with sodium sulfate (Na_2_SO_4_; pH 7.4) at same concentrations used as a negative control and incubated for 24 h in a humidified 5% CO_2_ incubator at 37 °C. Treatment with 100 μM of dexamethasone (D2915, Sigma-Aldrich) was used as a positive control. For inhibitor studies, cells were pre-treated for 1 h with 1 mM PFA or 10 nM pan-FGFR inhibitor PD1703744, before the addition of 4 mM sodium phosphate for 24 h.

### 4.11. C2C12 Cell Culture

C2C12 myoblasts (ATCC) were split and plated into 6-well plates (FB01297, Fisherbrand) or seeded onto chamber slides (354104, Falcon) and cultured in growth media (DMEM with 4.5% g/L glucose, L-glutamine, and sodium pyruvate; 10013CV, Corning), including 20% fetal bovine serum (FBS; 26140079, Gibco) and 1% penicillin/streptomycin (15140122, Gibco). When cells reached 80% confluency, growth media were replaced with differentiation media containing 2% horse serum (16050122, Gibco) instead of FBS to induce myoblast differentiation into mature myotubes. C2C12 myotubes were treated with FGF23 or Dex for 24 h, followed by RNA isolation. Furthermore, C2C12 myoblast proliferation was quantified by cell count. In a 6-well dish (FB012927, Fisherbrand), C2C12 myoblasts were treated for 24 h with 25 ng/mL of FGF23 or FGF2. After, media was removed, and cells were washed with PBS (14190144, Gibco). An amount of 200 μL of 0.05% trypsin-EDTA (25300054, Gibco) was added to cells for 5 min before neutralizing with 1 mL of media. A 1:1 mixture of cell suspension and Trypan Blue Solution (25900Cl, Corning) was analyzed by a hemocytometer to quantify cells (3120, Hausser Scientific, Horsham, PA, USA). Additionally, in 6-well plates, C2C12 myoblasts were treated for 24 h with 25 ng/mL of FGF23 or FGF2. After 22.5 h, 200 μL of AlamarBlue Cell Viability Reagent (DAL1024, Invitrogen) was added to each well and incubated for 1.5 h. After, 100 μL of media was transferred to a 96-well plate (FB012931, Fisherbrand), and a Synergy H1 microplate reader (BioTek, Winooski, VT, USA) was used to quantify absorbance at 570 nm and to calculate % cell growth between different treatment groups.

### 4.12. FGF Injections in Mice

Male C57BL/6J mice were injected via tail vein with 200 μL of saline or with 5 μg of FGF1 or FGF23 in 200 μL of saline. After 20 min, kidney and skeletal muscle tissue was harvested, and total protein was extracted for Western blot analysis.

### 4.13. Immunocytochemistry

Primary mouse myoblasts were plated into chamber slides (Falcon 354104) and grown to 80% confluency in growth media before changing to differentiation media. Myotubes were fixed using 2% paraformaldehyde and 4% sucrose in PBS for 5 min. 0.3% Triton X-100 (BP151-500, Fisher) was added for 10 min, followed by a 3 min PBS wash. Blocking buffer consisting of 2% fetal bovine serum (26140079, Gibco), 3% BSA, and 0.2% fish gelatine in PBS (900.033, Biotrend, Miramar Beach, FL, USA) was added to the cells for 30 min before the addition of anti-alpha actinin at 1:500 in blocking buffer for 60 min. Cells were washed 3× for 5 min with blocking buffer before the addition of the secondary antibodies at 1:500 in blocking buffer for 60 min. After, cells were washed 3× for 5 min with blocking buffer, Prolong Diamond Antifade Mountant with DAPI (P36962, Invitrogen) was added, and cells were covered with a microscope cover glass. Cell slides were set to dry overnight and stored at −80 °C. Cells were imaged on a Leica DMi8 microscope.

### 4.14. Image Analysis

Hindlimb quantification from MRI was quantified by area using ImageJ software v1.53k (NIH, Bethesda, MD, USA). Myofiber cross-sectional area of the anti-laminin-stained gastrocnemius sections was quantified using Cross Analyzer FIJI Plugin from ImageJ. The area of all myofibers within the gastrocnemius muscle was quantified and reported as averages per mouse. Percent area of fibrosis was quantified from 4–6 different fields of view images per gastrocnemius section, and the area of gray function on ImageJ was used to quantify area of fibrosis. Average length and area of 25 myotubes from 4–5 images taken per coverslip were quantified using ImageJ. A myotube is defined as a tube-like structure with more than 2 nuclei. Myotube length was measured from one defined end to the other defined end. Width was measured in myotube areas that did not contain nuclei.

### 4.15. Statistical Analysis

Data are expressed as arithmetic means ± SEM, and n represents the number of independent experiments. Organization of data, statistical significance analysis between samples, and graphing were performed using GraphPad Prism 10. Statistical analysis was made using unpaired Student *t*-test or ordinary one-way ANOVA and pair *t*-test where indicated. Only differences with *p* < 0.05 were considered statistically significant. *T*-tests, one-way ANOVAs, and two-way ANOVAs were conducted using Prism 10 and reported as means ± SEM.

## 5. Conclusions

Lowering systemic phosphate levels in CKD patients might protect the skeletal muscle and increase muscle mass and function, thereby improving overall physical performance. Furthermore, the intake of a phosphate-rich diet, for example, in the form of processed food, might cause skeletal muscle damage in healthy individuals over time.

## Figures and Tables

**Figure 1 ijms-25-09308-f001:**
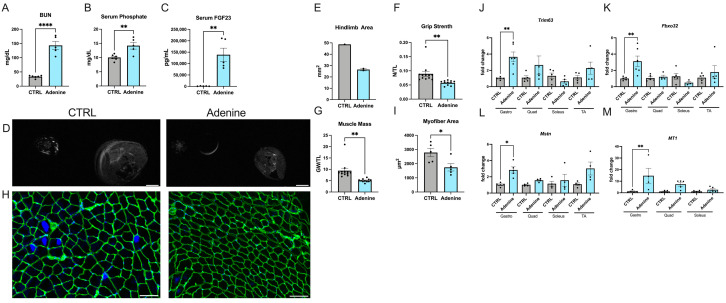
**Mice receiving an adenine-rich diet develop skeletal muscle atrophy.** We analyzed C57BL/6J mice that received an adenine-rich or control (CTRL) diet for 14 weeks. (**A**) Blood urea nitrogen (BUN) levels (*n* = 5–6; **** *p* < 0.0001), (**B**) serum phosphate levels (*n* = 5–6; ** *p* < 0.01), and (**C**) serum levels of FGF23 determined by ELISA (*n* = 5–6; ** *p* < 0.01). (**D**) Representative cross-sectional magnetic resonance images (MRI) from the hindlimb (scale bar = 2 mm). (**E**) Quantification of the hindlimb area in mm^2^ using MRI images (*n* = 1–2). (**F**) Grip strength in newtons (N) normalized to tibia length (TL) in mm (*n* = 11–12; ** *p* < 0.01). (**G**) Gastrocnemius weight (GW) in mg normalized to TL in mm (*n* = 11–12; ** *p* < 0.01). (**H**) Representative immunofluorescence images of gastrocnemius sections stained with anti-laminin to visualize cell borders and with DAPI to visualize cell nuclei (scale bar = 100 μm). (**I**) Quantification of the cross-sectional area of individual myofibers in μm^2^ based on anti-laminin immunostainings of gastrocnemius muscle sections (*n* = 5; * *p* < 0.05). (**J**–**M**) qRT-PCR expression analysis of gastrocnemius (gastro), quadricep (quad), soleus, and tibialis anterior (TA) muscles for *Tripartite motif containing 63* (*Trim63*), *F-box protein 32* (*Fbxo32*), *Myostatin* (*MSTN*), and *Metallothionein 1* (*MT1*) (*n* = 4–5; * *p* < 0.05, ** *p* < 0.01). Comparison between CTRL versus adenine mice was performed in an unpaired two-tailed *t*-test or a one-way ANOVA followed by a post-hoc Tukey’s test. All values are mean ± standard error of the mean (SEM).

**Figure 2 ijms-25-09308-f002:**
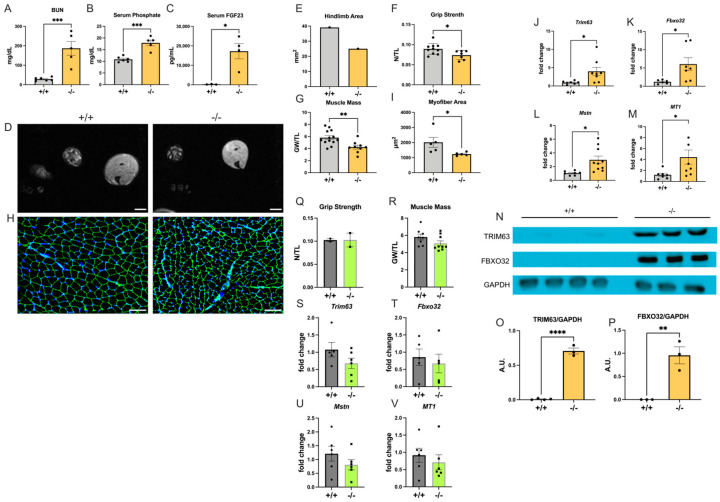
***Col4a3*^−/−^ mice develop skeletal muscle atrophy.** We analyzed mice with global deletion of *Collagen type IV alpha 3 chain* (−/−) and wildtype littermates (+/+) at 10 weeks (**A**–**P**) and 5 weeks (**Q**–**V**) of age. (**A**) Blood urea nitrogen (BUN) levels (*n* = 5–6; *** *p* < 0.001), (**B**) serum phosphate levels (*n* = 5–6; *** *p* < 0.001), and (**C**) serum levels of FGF23 determined by ELISA (*n* = 3–4; * *p* < 0.05). (**D**) Representative cross-sectional magnetic resonance images (MRI) from the hindlimb (scale bar = 2 mm). (**E**) Quantification of the hindlimb area in mm^2^ using MRI images (*n* = 1). (**F**) Grip strength in newtons (N) normalized to tibia length (TL) in mm (*n* = 7–10; * *p* < 0.05). (**G**) Gastrocnemius weight (GW) in mg normalized to TL in mm (*n* = 9–12; ** *p* < 0.01). (**H**) Representative immunofluorescence images of gastrocnemius muscle sections stained with anti-laminin to visualize cell borders and with DAPI to visualize cell nuclei (scale bar = 100 μm). (**I**) Quantification of the cross-sectional area of individual myofibers in μm^2^ based on anti-laminin immunostainings of gastrocnemius sections (*n* = 5; * *p* < 0.05). (**J**–**M**) qRT-PCR expression analysis of the gastrocnemius for *Tripartite motif containing 63* (*Trim63*), *F-box protein 32* (*Fbxo32*), *Myostatin* (*MSTN*), and *Metallothionein 1* (*MT1*) (*n* = 6–10; * *p* < 0.05). (**N**) Representative images of Western blots to analyze the protein expression of TRIM63 and FBXO32 in isolated gastrocnemius tissue. GAPDH serves as a loading control. (**O**,**P**) Quantification of Western blot signals of TRIM63 and FBXO32 by densitometry (*n* = 3–4; ** *p* < 0.01, **** *p* < 0.0001). (**Q**) Grip strength of 5-week-old mice in N normalized to TL in mm (*n* = 2). (**R**) GW in mg normalized to TL in mm (*n* = 8–9). (**S**–**V**) qRT-PCR expression analysis of the gastrocnemius for *Trim63*, *Fbxo32*, *MSTN*, and *MT1* (*n* = 5–6). Comparison between *Col4a3^+^*^/*+*^ versus *Col4a3^−^*^/*−*^ mice was performed in an unpaired two-tailed *t*-test. All values are mean ± standard error of the mean (SEM).

**Figure 3 ijms-25-09308-f003:**
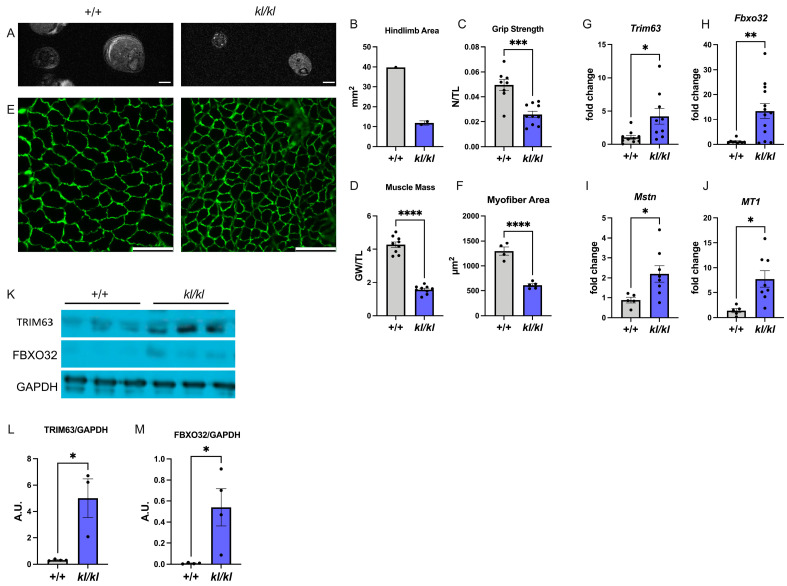
***Klotho*-deficient mice develop skeletal muscle atrophy.** Homozygous *klotho*-deficient mice (*kl*/*kl*) and wildtype littermates (+/+) were analyzed at 8 weeks of age. (**A**) Representative cross-sectional magnetic resonance images (MRI) from the hindlimb (scale bar = 2 mm). (**B**) Quantification of the hindlimb area in mm^2^ using MRI images (*n* = 1–2). (**C**) Grip strength in newtons (N) normalized to tibia length (TL) in mm (*n* = 8–10; *** *p* < 0.001). (**D**) Gastrocnemius weight (GW) in mg normalized to TL in mm (*n* = 9; **** *p* < 0.0001). (**E**) Representative immunofluorescence images of gastrocnemius muscle sections stained with anti-laminin to visualize cell borders and with DAPI to visualize cell nuclei (scale bar = 100 μm). (**F**) Quantification of the cross-sectional area of individual myofibers in μm^2^ based on anti-laminin immunostainings of gastrocnemius sections (*n* = 4–5; **** *p* < 0.0001). (**G**–**J**) qRT-PCR expression analysis of the gastrocnemius for *Tripartite motif containing 63* (*Trim63*), *F-box protein 32* (*Fbxo32*), *Myostatin* (*MSTN*), and *Metallothionein 1* (*MT1*) (*n* = 5–13; * *p* < 0.05, ** *p* < 0.01). (**K**) Representative images of Western blots to analyze the protein expression of TRIM63 and FBXO32 in isolated gastrocnemius tissue. GAPDH serves as a loading control. (**L,M**) Quantification of Western blot signals of TRIM63 and FBXO32 by densitometry (*n* = 3–4; * *p* < 0.05). Comparison between +/+ versus *kl*/*kl* mice was performed in an unpaired two-tailed *t*-test. All values are mean ± standard error of the mean (SEM).

**Figure 4 ijms-25-09308-f004:**
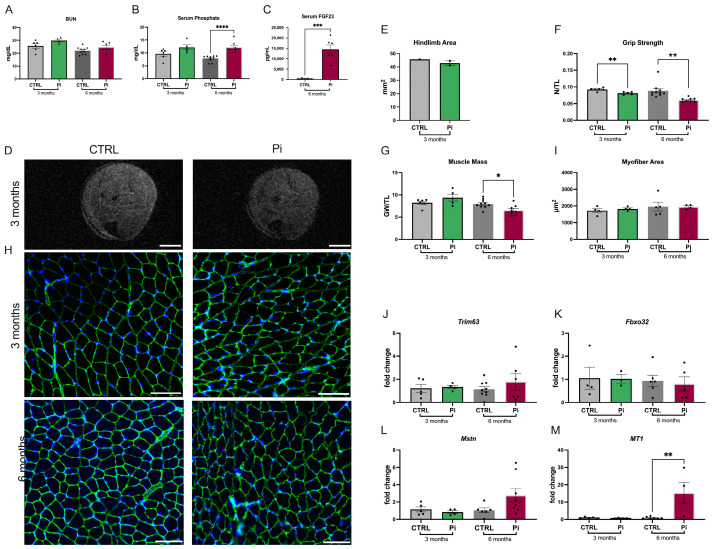
**Mice receiving a high-phosphate diet develop reduced skeletal muscle mass and function but not atrophy.** We analyzed C57BL/6J mice that received a 3% phosphate (Pi) or control (CTRL) diet for 3 or 6 months. (**A**) Blood urea nitrogen (BUN) levels (*n* = 4–10), (**B**) serum phosphate levels (*n* = 5–10; **** *p* < 0.0001), and (**C**) serum levels of FGF23 determined by ELISA (*n* = 5–6; *** *p* < 0.001). (**D**) Representative cross-sectional magnetic resonance images (MRI) from the hindlimb (scale bar = 2 mm). (**E**) Quantification of the hindlimb area in mm^2^ using MRI images (*n* = 1–2). (**F**) Grip strength in newtons (N) normalized to tibia length (TL) in mm (*n* = 5–10; ** *p* < 0.01). (**G**) Gastrocnemius weight (GW) in mg normalized to TL in mm (*n* = 5–10; * *p* < 0.05). (**H**) Representative immunofluorescence images of gastrocnemius muscle sections stained with anti-laminin to visualize cell borders and with DAPI to visualize cell nuclei (scale bar = 100 μm). (**I**) Quantification of the cross-sectional area of individual myofibers in μm^2^ based on anti-laminin immunostainings of gastrocnemius sections (*n* = 4–5). (**J**–**M**) qRT-PCR expression analysis of the gastrocnemius for *Tripartite motif containing 63* (*Trim63*), *F-box protein 32* (*Fbxo32*), *Myostatin* (*MSTN*), and *Metallothionein 1* (*MT1*) (*n* = 3–8; ** *p* < 0.01). Comparison between 3-month CTRL and Pi or between 6-month CTRL and Pi was performed using an unpaired two-tailed *t*-test or a one-way ANOVA followed by a post-hoc Tukey’s test. All values are mean ± standard error of the mean (SEM).

**Figure 5 ijms-25-09308-f005:**
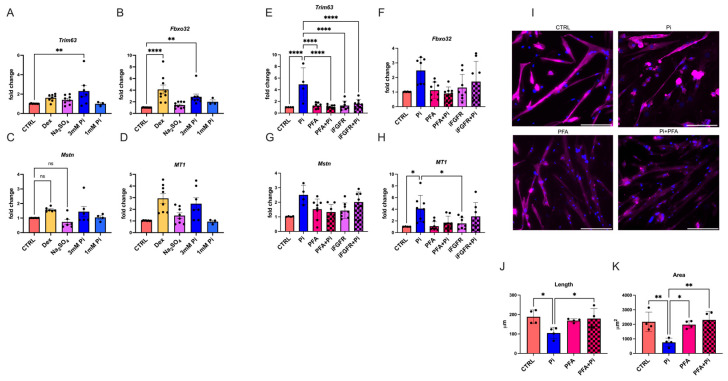
**Phosphate elevations induce atrophy in primary mouse myotubes.** Myotubes were isolated from skeletal muscle tissue of adult C57BL/6J mice and cultured in the presence of normal medium (CTRL),100 μM dexamethasone (Dex), 3 mM Na_2_SO_4_, or 1 or 3mM phosphate (Pi) for 24 h. (**A**–**D**) qRT-PCR expression analysis for *Tripartite motif containing 63* (*Trim63*), *F-box protein 32* (*Fbxo32*), *Myostatin* (*MSTN*), and *Metallothionein 1* (*MT1*) (*n* = 3–10; * *p* < 0.05, ** *p* < 0.01, **** *p* < 0.0001). (**E**–**H**) Primary mouse myotubes were cultured in normal medium (CTRL) or in 4 mM Pi for 24 h. Some cells were pre-treated with 1mM phosphonoformic acid (PFA) or 20 ng/mL of a pan-FGFR inhibitor (iFGFR) for 1 h. qRT-PCR expression analysis for *Trim63*, *Fbxo32*, *MSTN*, and *MT1* (*n* = 4–8; * *p* < 0.05, **** *p* < 0.0001). (**I**) Representative images of primary mouse myotubes stained with anti-α-actinin to visualize myotubes and with DAPI to visualize nuclei (scale bar = 100 μm). (**J**) Average length of myotubes based on α-actinin staining and quantified from 4–5 different fields of view per separate cell isolation (*n* = 4 mice; * *p* < 0.05). (**K**) Average area of myotubes based on α-actinin staining and quantified from 4–5 different fields of view per separate cell isolation (*n* = 4 mice; * *p* < 0.05, ** *p* < 0.01). Comparison between cell treatment groups was performed using a one-way ANOVA followed by a post-hoc Tukey’s test. All values are mean ± standard error of the mean (SEM).

## Data Availability

The data on which this article are based are available in the article and in its online Appendix A.

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
