# Peer review of "Hyperphosphatemia Contributes to Skeletal Muscle Atrophy in Mice"

_ijms, 2024, doi:10.3390/ijms25179308_

Round 1

Reviewer 1 Report

Comments and Suggestions for Authors

General:

This study by Heitman et al is interesting and really well done. They use models of endogenous and exogenous hyperphosphatemia to address the question whether and what type of muscle damage is directly or indirectly caused by hyperphosphatemia in CKD. The study is informative, the data is of good quality, and extensive in vivo experiments were performed. However, some conclusions are overstated, some data is missing, some discrepancies with existing literature need to be addressed, and the discussion is too long. Specific comments below:

Specific Comments:

1) The authors claim that in the Alport CKD model low Phos diet prevents muscle pathology (this is key data allowing the authors to attribute their phenotypes to the effect of hyperphosphatemia) but this data appears to be missing. Fig 2J-M shows a few select transcripts but generally no actual biological conclusions can ever be drawn just from RTPCR data as transcripts are highly volatile and do not have a lot of meaning. I do not find any data on actual muscle pathology (function, structure, histology).

2) The abstract is lengthy and hard to read. Conclusions are overstated. We really do not know what this could mean for humans as actual people are different and not studied here. It is generally better to stick to what the data shows and there already are enough strong conclusions as the authors have done a lot of nice experiments. I would suggest to remove the last four sentences of the abstract (including the fibrosis, inflammation statement) and instead I would say something along the lines of: “We conclude that in experimental CKD models hyperphosphatemia is sufficient to induce muscular dysfunction but there are additional CKD-specific mechanisms”.

3) “determine” appears four time in the abstract. One could maybe replace with another word in some instances.

4) How do the authors know that in any of their models muscle pathology is actually related to or caused by hyperphosphatemia? Klotho is expressed in the muscle with a known myogenic role there (PMID: 37864311) and it looks like the authors used global Klotho deletion. Similarly, the muscle pathology in the Adenine diet CKD model was recently shown to be due to a temporary fully recoverable shift in energy metabolism and caution was urged to interpret muscle function in experimental ckd models (PMID: 38509307). Please address.

5) Figure 1H: no major difference in myofiber area based on images and resolution provided. Please indicate what was quantified in Figure 1I. Based on bar graph ~40% reduction with adenine but this does not appear to be reflected in the image.

6) Figure 5: very strong conclusions on phosphate affecting muscle atrophy and differentiation are drawn largely based on transcripts in a cell culture model. This just cannot be done as transcripts are the weakest form of data. At a minimum, please measure myotube diameter (please add a representative high-res image of these myotubes to the figure) and MyHC protein content to add biologically meaningful data to this figure.

7) Figure 6: same issue. Also, the resolution of images in 6I is really low. Please take a high res image, and zoom in on a myotube indicating with annotations where length and area measurements were taken, showing a nice difference that reflects the quantification.

8) The discussion is significantly too long, almost 4 full pages. The part on clinical implications can largely go out. This study has little clinical implications which is fine. Even the existing human clinical evidence for Mineral-bone-muscle disease or phosphate interventions in CKD is absolutely terrible so the entire discussion on clinical implications of a mouse study seems far-fetched and distracts from the really interesting and strong experimental biology that this paper contains.

Author Response

Reviewer: This study by Heitman et al is interesting and really well done. They use models of endogenous and exogenous hyperphosphatemia to address the question whether and what type of muscle damage is directly or indirectly caused by hyperphosphatemia in CKD. The study is informative, the data is of good quality, and extensive in vivo experiments were performed. However, some conclusions are overstated, some data is missing, some discrepancies with existing literature need to be addressed, and the discussion is too long.

Response: We thank the Reviewer for the positive feedback, and we have addressed the comments and suggestions for modifications in a point-by-point fashion below.

Reviewer: The authors claim that in the Alport CKD model low Phos diet prevents muscle pathology (this is key data allowing the authors to attribute their phenotypes to the effect of hyperphosphatemia) but this data appears to be missing. Fig 2J-M shows a few select transcripts but generally no actual biological conclusions can ever be drawn just from RTPCR data as transcripts are highly volatile and do not have a lot of meaning. I do not find any data on actual muscle pathology (function, structure, histology).

Response: We have published the effect of the low-phosphate diet on skeletal muscle structure and function in the Alport mouse model before in eLife (see Figure 5 in PMID 35302487), where we show improvements in grip strength, muscle mass and histology. In the current manuscript, we only wanted to add the changes that we observed in some additional molecular markers which indicate reduced skeletal muscle atrophy in Alport mice following a low-phosphate diet. However, since these findings do not add significantly novel insights to the already published data, we have decided to delete the qPCR analysis from Alport mice on low-phosphate diet from Figure 2J-M in the revised manuscript. Compared to our study in eLife the novelty of the data presented here in Figure 2, where we study new groups of Alport mice and controls, is a more detailed analysis of the skeletal muscle phenotype, including reductions in the area of individual myofibers, which is the key readout for atrophy. Furthermore, for the first time we show here that younger Alport mice without kidney injury and hyperphosphatemia do not develop skeletal muscle atrophy, suggesting that in this animal model skeletal muscle injury only occurs in the presence of kidney injury and hyperphosphatemia.

Reviewer: The abstract is lengthy and hard to read. Conclusions are overstated. We really do not know what this could mean for humans as actual people are different and not studied here. It is generally better to stick to what the data shows and there already are enough strong conclusions as the authors have done a lot of nice experiments. I would suggest to remove the last four sentences of the abstract (including the fibrosis, inflammation statement) and instead I would say something along the lines of: “We conclude that in experimental CKD models hyperphosphatemia is sufficient to induce muscular dysfunction but there are additional CKD-specific mechanisms”.

Response: We agree with the Reviewer, and we have edited and shortened the Abstract accordingly.

Reviewer: “determine” appears four time in the abstract. One could maybe replace with another word in some instances.

Response: We agree, and we have replaced the word “determine” in some of the sentences.

Reviewer: How do the authors know that in any of their models muscle pathology is actually related to or caused by hyperphosphatemia? Klotho is expressed in the muscle with a known myogenic role there (PMID: 37864311) and it looks like the authors used global Klotho deletion.

Response: We agree with the Reviewer that klotho might have direct protective effects on skeletal muscle tissue and that mice with a global loss of klotho develop skeletal muscle injury based on the absence of these direct effects. However, it remains unclear if klotho is expressed in skeletal muscle tissue (not every published study could detect klotho in skeletal muscle) and/or if circulating soluble klotho can target skeletal muscle. We have included this potential explanation in the revised Discussion. We also want to point out that others have shown that the deletion of the renal phosphate transporter NaPi-2a in klotho-deficient mice reduces serum phosphate levels and protects from skeletal muscle atrophy, suggesting that in these mice skeletal muscle injury is not directly caused by the absence of klotho, but indirectly by other pathologic changes that occur in the absence of klotho, such as hyperphosphatemia. This point is also discussed.

Reviewer: Similarly, the muscle pathology in the Adenine diet CKD model was recently shown to be due to a temporary fully recoverable shift in energy metabolism and caution was urged to interpret muscle function in experimental ckd models (PMID: 38509307). Please address.

Response: We also read the only recently published study by Lair and colleagues with great interest, showing that mice on an adenine-rich diet develop reduced skeletal muscle mass due to a temporary and reversible shift in energy metabolism (PMID: 38509307). Furthermore, this study found that following 5/6 nephrectomy mice show reductions in muscle mass early on due to reduced food intake and a drop in overall body mass; however, later on mice recover from their pathologic changes in skeletal muscle tissue. Both models suggest that kidney injury per se, which in these mice does not improve over time, might not be a driver of skeletal muscle damage in CKD. However, we want to point out that more than 10 other studies conducted in 5/6 nephrectomized mice and rats have reported skeletal muscle damage, including atrophy, as we have summarized and discussed in a recent review article in IJMS (PMID: 38791164). Furthermore, 6 other studies have analyzed skeletal muscle in adenine mice with various outcomes (also reviewed in PMID: 38791164). While the explanation for the discrepancies between the paper by Lair et al and the other published work and our own work is not clear, they might be based on differences in the precise design of the model and study. For example, Lair et al administered the adenine-rich diet for 3 weeks, removed it for 3 days, and then provide it again for 1 week, and then analyzed the skeletal muscle phenotype several weeks after the adenine diet has been removed. We and others in the field have conducted continues adenine feeding studies for 14 weeks (6 weeks of a 0.2% adenine diet, followed by 2 weeks of a 0.15% adenine diet, and an additional 6 weeks of 0.2% adenine diet). It seems that based on the massive weight loss that Lair et al detected in their adenine mice from early on and the associated ethical reasons, they had to significantly modify their feeding protocol. We have not seen these severe changes in our adenine mice, and it brings up the point whether for Lair et al the adenine diet had toxic effects, other than inducing kidney injury. Again, even if it remains unclear why these studies have different outcomes, it is important to report all of these findings to demonstrate that animal studies need to be taken with caution when interpreting pathologies and that nuances in the design of the model and study might effect outcomes. The key question that remains is which models reflect the human pathology the best, which will be challenging to answer as the characterization of CKD-associated skeletal muscle injury in patients is even less detailed and not very advanced. We have included the new study by Lair and colleagues in our revised Discussion.

Reviewer: Figure 1H: no major difference in myofiber area based on images and resolution provided. Please indicate what was quantified in Figure 1I. Based on bar graph ~40% reduction with adenine but this does not appear to be reflected in the image.

Response: We have replaced the anti-laminin immunofluorescence image for the adenine mouse model in Figure 1H with a more representative image. For the analysis of myofiber cross-sectional area in tissue, anti-laminin stained gastrocnemius sections were quantified using Cross Analyzer FIJI Plugin from ImageJ. The area of all myofibers within the gastrocnemius muscle were quantified and reported as averages per mouse. We have included this information in the Methods section of the revised manuscript.

Reviewer: Figure 5: very strong conclusions on phosphate affecting muscle atrophy and differentiation are drawn largely based on transcripts in a cell culture model. This just cannot be done as transcripts are the weakest form of data. At a minimum, please measure myotube diameter (please add a representative high-res image of these myotubes to the figure) and MyHC protein content to add biologically meaningful data to this figure.

Response: We have measured length and area of isolated primary mouse myotubes (Figure 5I-K). We describe the measurements in the Methods section of the revised manuscript. Since the primary cell culture model is more relevant and representative of myofibers within mouse tissue, we decided to only show primary cells and take out the data derived from the C2C12 cell line, where we only studied atrophy on the level of changes in molecular markers but not in cell area. Therefore, the previous Figure 5 has been deleted. We agree with the Reviewer that the potential inhibitory effect of elevated phosphate on differentiation, as studied in C2C12 myoblasts (previous Figure 5F-H), needs additional readouts to indicate the absence or presence of differentiated myotubes. However, we believe that studying myoblast-to-myotube differentiation is not relevant for the current manuscript, since in our mouse models we focus on the effects of high phosphate on adult skeletal muscle, and not on skeletal muscle during development or following primary muscle injury. While it would be interesting to determine if elevated phosphate affects skeletal muscle regeneration in CKD, this is beyond the scope of the current manuscript. Therefore, we decided to delete the analysis of C2C12 cell differentiation.

Reviewer: Figure 6: same issue. Also, the resolution of images in 6I is really low. Please take a high res image, and zoom in on a myotube indicating with annotations where length and area measurements were taken, showing a nice difference that reflects the quantification.

Response: We agree with the Reviewer. We have revised this figure (now Figure 5I), and we present images that were taken at higher magnification showing individual myotubes.

Reviewer: The discussion is significantly too long, almost 4 full pages. The part on clinical implications can largely go out. This study has little clinical implications which is fine. Even the existing human clinical evidence for Mineral-bone-muscle disease or phosphate interventions in CKD is absolutely terrible so the entire discussion on clinical implications of a mouse study seems far-fetched and distracts from the really interesting and strong experimental biology that this paper contains.

Response: We agree with the Reviewer, and we have significantly edited and shortened the Discussion, including the section discussing potential clinical implications, to stay closer connected to our actual experimental findings.

Reviewer 2 Report

Comments and Suggestions for Authors

Heitman et al investigated the role of hyperphosphatemia for skeletal muscle atrophy in mice. They used four mouse models and C2C12 myotube culture to tease out the role of hyperphosphatemia with versus without the additional impact of CKD. Rationale of their current study is clearly explained in the section of discussion. Relevant methodologies are provided, and results are presented in a logical way. In addition, their results largely supported their notion, i.e., hyperphosphatemia is an important cause of muscle atrophy in the setting or CKD or by itself alone.

I have the following concerns.

This is a study, kind of repetitive of what have been published and lack of novelty. The detailed molecular pathways underlying hyperphosphatemia and skeletal muscle atrophy have been discussed by three recent publications. The authors need to highlight what set apart (this current study) from those recently published studies (below). Specifically, the data on Klotho-deficient mice has not been published. The rest of the results for this manuscript is repetitive of what have been published by the following three articles. High phosphate has been shown to induce skeletal muscle atrophy and suppresses myogenic differentiation by increasing oxidative stress and activating Nrf2 signaling (Chung et al. Aging 2020;12:No.21,page 21446). Two additional articles by this group of established investigators have been published. In FASEB, they have clearly detailed the putative role of hyperphosphatemia for muscle atrophy with and without the setting of CKD. They also published in Elife 2022 about the association between hyperphosphatemia/inflammation/anemia/skeletal muscle which is independent of FGF23-FGFR4 signaling. 

Some other minor issues. 1) For an original article, a total of 198 references is excess. This is not a literature review. 2) For most of the figures, the font size of the data set and the labeling is too small. It is hard to read. 3) Some typo, e.g., line 302, C1C12 myotube (should be C2C12).

Comments on the Quality of English Language

No further comments, minor editing issue

Author Response

Reviewer: Heitman et al investigated the role of hyperphosphatemia for skeletal muscle atrophy in mice. They used four mouse models and C2C12 myotube culture to tease out the role of hyperphosphatemia with versus without the additional impact of CKD. Rationale of their current study is clearly explained in the section of discussion. Relevant methodologies are provided, and results are presented in a logical way. In addition, their results largely supported their notion, i.e., hyperphosphatemia is an important cause of muscle atrophy in the setting or CKD or by itself alone.

Response: We thank the Reviewer for the positive feedback.

Reviewer: This is a study, kind of repetitive of what have been published and lack of novelty. The detailed molecular pathways underlying hyperphosphatemia and skeletal muscle atrophy have been discussed by three recent publications. The authors need to highlight what set apart (this current study) from those recently published studies (below). Specifically, the data on Klotho-deficient mice has not been published. The rest of the results for this manuscript is repetitive of what have been published by the following three articles. High phosphate has been shown to induce skeletal muscle atrophy and suppresses myogenic differentiation by increasing oxidative stress and activating Nrf2 signaling (Chung et al. Aging 2020;12:No.21,page 21446). Two additional articles by this group of established investigators have been published. In FASEB, they have clearly detailed the putative role of hyperphosphatemia for muscle atrophy with and without the setting of CKD. They also published in Elife 2022 about the association between hyperphosphatemia/inflammation/anemia/skeletal muscle which is independent of FGF23-FGFR4 signaling.

Response: We want to point out that our publication in FASEB is an abstract from our group for a presentation at the 2021 Experimental Biology Meeting. Compared to studies published by others (which we recently presented and discussed in a review article in IJMS; PMID: 38791164) and our own study published in eLife in 2022 (PMID: 35302487), the current manuscript provides several novel aspects:

In all four mouse models, we provide a more detailed analysis of the skeletal muscle phenotype, including changes in molecular markers, histology, muscle mass and muscle function, including reductions in the area of individual myofibers, which is the key readout for atrophy. Furthermore, we also analyze muscle inflammation and fibrosis, and for some of the molecular markers we study different types of skeletal muscle, not just the gastrocnemius muscle that is usually used in most of the other studies. Based on this detailed analysis we provide the novel insight that different types of skeletal muscle respond differently to hyperphosphatemia. This finding highlights the need for targeted investigations into how various muscle types are affected by CKD and hyperphosphatemia. We found that in our two CKD models, muscle atrophy is not accompanied by interstitial fibrosis or inflammation, suggesting that the reduction in muscle function is primarily due to atrophy rather than changes in muscle quality, and indicating that fibrosis and inflammation might not be universal features of CKD-associated muscle atrophy. In a recent review article in IJMS, we pointed out the important gaps in the field of CKD-associated sarcopenia, including the gaps in animal studies (PMID: 38791164), and with the current research manuscript we aim to fill some of these gaps. The significance for moving forward with these characterizations in animal models of CKD is high, so that one day researchers can identify novel mechanisms of sarcopenia and thereby novel drug targets, which cannot be done by human studies.

For the first time we show that younger Alport mice without kidney injury and hyperphosphatemia do not develop skeletal muscle atrophy, suggesting that in this animal model skeletal muscle injury only occurs in the presence of kidney injury and hyperphosphatemia.

The analysis of skeletal muscle tissue in mice receiving a high-phosphate diet for 20 weeks, as published by others(PMID: 33136552), is not as detailed as our analysis presented here. Furthermore, for the first time we conduct a time course, and we distinguish between 3 versus 6 months of high-phosphate diet. We find that in this model changes in skeletal muscle tissue are not severe, but that time might have an impact, as we detected changes on molecular level after the longer feeding period.

We present pathologic actions of elevated phosphate levels on cultured myotubes. While another study found that phosphate inhibits the differentiation of C2C12 myoblasts (including impaired mitochondrial function and elevated ROS; PMID: 33136552), our study is the first to detect effects of phosphate on atrophy, which is the topic of our study and a major skeletal muscle pathology in CKD. Furthermore, we analyze isolated primary mouse myotubes, which we think is a more relevant cell culture model for studying myofibers compared to the C2C12 cell line. Our in vitro study suggests for the first time that at high levels phosphate can directly induce skeletal muscle atrophy.

We also read the only recently published study by Lair and colleagues with great interest, showing that mice on an adenine-rich diet develop reduced skeletal muscle mass due to a temporary and reversible shift in energy metabolism (PMID: 38509307). Furthermore, this study found that following 5/6 nephrectomy mice show reductions in muscle mass early on due to reduced food intake and a drop in overall body mass; however, later on mice recover from their pathologic changes in skeletal muscle tissue. Both models suggest that kidney injury per se, which in these mice does not improve over time, might not be a driver of skeletal muscle damage in CKD. However, we want to point out that more than 10 other studies conducted in 5/6 nephrectomized mice and rats have reported skeletal muscle damage, including atrophy, as we have summarized and discussed in a recent review article in IJMS (PMID: 38791164). Furthermore, 6 other studies have analyzed skeletal muscle in adenine mice with various outcomes (also reviewed in PMID: 38791164). While the explanation for the discrepancies between the paper by Lair et al and the other published work and our own work is not clear, they might be based on differences in the precise design of the model and study. For example, Lair et al administered the adenine-rich diet for 3 weeks, removed it for 3 days, and then provide it again for 1 week, and then analyzed the skeletal muscle phenotype several weeks after the adenine diet has been removed. We and others in the field have conducted continues adenine feeding studies for 14 weeks (6 weeks of a 0.2% adenine diet, followed by 2 weeks of a 0.15% adenine diet, and an additional 6 weeks of 0.2% adenine diet). It seems that based on the massive weight loss that Lair et al detected in their adenine mice from early on and the associated ethical reasons, they had to significantly modify their feeding protocol. We have not seen these severe changes in our adenine mice, and it brings up the point whether for Lair et al the adenine diet had toxic effects, other than inducing kidney injury. Again, even if it remains unclear why these studies have different outcomes, it is important to report all of these findings to demonstrate that animal studies need to be taken with caution when interpreting pathologies and that nuances in the design of the model and study might effect outcomes. The key question that remains is which models reflect the human pathology the best, which will be challenging to answer as the characterization of CKD-associated skeletal muscle injury in patients is even less detailed and not very advanced.

Reviewer: For an original article, a total of 198 references is excess. This is not a literature review.

Response: We agree with the Reviewer, and we have edited and shortened the Introduction and Discussion sections, which resulted in a significant reduction in the number of references.

Reviewer: For most of the figures, the font size of the data set and the labeling is too small. It is hard to read.

Response: We agree, and we have increased the font size for the labeling of x and y axis in all figures.

Reviewer: Some typo, e.g., line 302, C1C12 myotube (should be C2C12).

Response: This has been changed.

Round 2

Reviewer 1 Report

Comments and Suggestions for Authors

I thank the authors for appropriately addressing all my comments. Particularly, I am grateful for the new data, the explanations, and for pointing me to the pertinent literature. I learned a lot and the manuscript is suitable for publication. Congratulations on an excellent study!